# Creation of Cellulolytic Communities of Soil Microorganisms—A Search for Optimal Approaches

**DOI:** 10.3390/microorganisms12112276

**Published:** 2024-11-09

**Authors:** Aleksei O. Zverev, Anastasiia K. Kimeklis, Olga V. Orlova, Tatiana O. Lisina, Arina A. Kichko, Alexandr G. Pinaev, Alla L. Lapidus, Evgeny V. Abakumov, Evgeny E. Andronov

**Affiliations:** 1All-Russian Research Institute for Agricultural Microbiology (ARRIAM), 3 Podbelsky Chaussee, 196608 Saint Petersburg, Russia; 2Dokuchaev Soil Science Institute, Pyzhyovskiy Lane 7, 119017 Moscow, Russia; 3Department of Applied Ecology, Faculty of Biology, Saint Petersburg State University, 199034 Saint Petersburg, Russia; 4Independent Researcher, 125493 Saint Petersburg, Russia

**Keywords:** amplicon sequencing, cellulolytic microorganisms, cellulose degradation, hemp biodegradation, straw biodegradation

## Abstract

For the targeted selection of microbial communities that provide cellulose degradation, soil samples containing cellulolytic microorganisms and specific plant residues as a substrate can be used. The details of this process have not been studied: in particular, whether the use of different soils determines the varying efficiency of communities; whether these established cellulolytic communities will have substrate specificity, and other factors. To answer these questions, four soil microbial communities with different cellulolytic activity (Podzol and the soil of Chernevaya taiga) and substrates (oat straw and hemp shives) with different levels of cellulose availability were used, followed by trained communities that were tested on botrooth substrates (in all possible combinations). Based on the analysis of the taxonomic structure of all communities and their efficiency across all substrates (decomposition level, carbon, and nitrogen content), it was shown that the most important taxa of all trained microbial cellulolytic communities are recruited from secondary soil taxa. The original soil does not affect the efficiency of cellulose decomposition: both soils produce equally active communities. Unexpectedly, the resulting communities trained on oats were more effective on hemp than the communities trained on hemp. In general, the usage of pre-trained microbial communities increases the efficiency of decomposition.

## 1. Introduction

The decomposition and mineralization of organic residues is one of the most important functions of soils. Decomposition allows for the rapid replenishment of the pool of available carbon in the soil, providing stable conditions and broad opportunities for plant growth and development. One of the most common substrates for such decomposition is cellulose. Cellulose decomposition processes in soil are carried out by cellulolytic microorganisms, represented mainly by bacteria and fungi [1,2].

The decomposition of plant residues is one of the most important service functions of soil microbiota [3]. These processes have been going on for more than 2 billion years, leading to the formation of highly efficient communities. Diverse microbial associations are not only formed by soils of different types, but are also formed under the influence of various factors: the decomposition of plant residues in different climatic conditions, on plant residues of different origins, etc. [2]. All these processes, lasting over billions of years, leave traces in the structure of microbial associations and provide variability in decomposition processes.

Cellulose degradation processes are regularly a subject of interest in agriculture, where there is an ongoing issue related to the relatively rapid utilization of large quantities of plant residues [4]. The direct cultivation of cellulolytic microorganisms seems to be the most obvious approach, and indeed many studies have been devoted to isolating and studying the metabolic activity of individual bacterial strains [5,6]. This approach, however, comes with several challenges; the main ones are the difficulty of culturing certain monocultures, the general decrease in enzyme efficiency in monocultures, and the very high specificity of the resulting enzymes [7]. On the contrary, the use of natural communities allows the use of a wide range of substrates and significantly increases the efficiency of biodegradation processes of cellulose-containing substrates.

Due to their nature as a universal source, soil communities themselves are an excellent source of cellulose-degrading microbial communities [8,9]. The destructive function of the soil community can be enhanced by directed selection. Using soil as a source of cellulose-degrading associations, and specific substrates as selective media, it is possible to “breed” communities that are highly effective in decomposing specific types of plant residues.

However, the details of setting up such a training process are not fully clear. What soils are the best choice for producing effective communities? Local soils are better adapted to local conditions, but their basic cellulolytic potential is often lower than the one of special, selected soils. Will soils with active microbial processes be a better source of cellulolytic microorganisms than poor soils? Do differences in soil microbial diversity levels play a role? In other words, is the final cellulolytic potential acquired during training determined by soil characteristics, or, according to Baas Becking’s principle that “Everything is everywhere, but the environment selects”, can active microbial associations be recruited from any soil?

The second important question about the usage of different substrates for this selection of cellulolytic associations is the following: does substrate-specificity of communities arise in the selection process? That is, if we need to obtain a community that efficiently degrades a specific substrate, should we use that substrate in selection? Is substrate-specificity a characteristic of cellulolytic communities in general? The last point is particularly important because different types of plant residues are enriched with structural components of different natures—some are dominated by easily degradable forms of cellulose, while others are enriched with difficult-to-degrade lignin derivatives.

In the present study, the diversity of these aspects is reduced to two factors:-Podzolic soil and the soil of Chernevaya taiga were used as contrasting soils. The soils differ both in initial cellulolytic activity, nutrient content, and species richness of prokaryotes. Relatively poor in nutrients, Podzolic soil is one of the common soils in the northern part of Russia and can be found easily. In contrast, relatively rich Chernevaya taiga soil is a site-specific soil from the dark forests of Siberia.-Oat and hemp straw were used as substrates with different recalcitrance characteristics. Hemp straw contains a high amount of lignin, which is a difficult-to-degrade substrate and accumulates as a side-product of the manufacturing of hemp threads, rope, and fabrics. The main ways of utilizing hemp straw are the decomposition of cellulose residuals and using the resulting biomass as a fertilizer. Nowadays, the production of biofuels from hemp straw is unreasonable due to low yields and the high complexity of its degradation processes. Oat straw contains less lignin and can be decomposed easily. It also accumulates as an agricultural residual and can be used in a variety of ways, from fertilizer production to biofuel production.

To determine the trends and aspects of the formation of cellulose-degrading communities, a microbiome enrichment experiment with two contrasting soils and two substrates of different availability levels was set up. The overall goal of the research was to determine the differences in the microbial composition and cellulolytic activity of acquired communities. The goals in particular were to (i) describe these communities in terms of ecological microbiology, (ii) determine the origin of the most abundant taxa in these communities, (iii) estimate the efficiency of these communities on both substrates, and (iv) find out whether there is any substrate-specificity in the efficiency of substrate decomposition.

## 2. Materials and Methods

Two types of soil were used as a source of cellulose-degrading microorganisms: the soil of Chernevaya taiga (Tomsk region, Russia) [10,11] and sod-podzolic soil (Volgograd region, Russia) [12,13]. Both soils were sieved on a 5 cm sieve to remove roots and large debris; moisture content was adjusted to 60% of full capacity. The difference in the initial cellulolytic activity was demonstrated using the decomposition of a paper filter on a Petri dish (Figure 1).

For the creation of pre-trained microbial communities (referred to later as primers), which are used to initiate and carry out the decomposition of the substrate (training phase, or phase I), the principle of accumulation culture was used. To start the primer, 300 g of substrate and 500 g of soil were placed in a 1-L vessel. Cultivation lasted for 1 year and 2 months, with the periodic addition of 5% cellulose substrate and incubation at a temperature of 28 °C and humidity of 60% of full moisture capacity. Oat straw and hemp straw were used as substrates for decomposition. Moisture was adjusted by weight loss every 7 days alongside intense mixing. Moisture content, ash content, and total carbon content were measured and samples were taken for microbial analysis.

To investigate the specificity of the communities (testing phase, or phase II), prepared primers were added to both available substrates (oat straw and hemp straw) at the ratio of 10 g of dry substrate, 1 g of primer, and 30 mL of water. Cultivation was carried out for 110 days at 28 °C and weekly control of humidity. At the end of the experiment, the moisture content, ash content, residue of the applied substrate (determined by cellulose weight loss), nitrate content, water-soluble carbon content, and microbial analysis were measured and samples were taken for microbial analysis. The overall design of the experiment is summarized in Figure 2.

DNA was isolated from all samples in 4 replicas using the RIAM protocol [14]; the concentration and purity of DNA isolation were checked via electrophoretic separation in 1% agarose gel. Sequencing was performed on an Illumina MiSEQ (San Diego, CA, USA) using specific primers to the v3–v4 variable region of the 16S rRNA gene: f515 (GTGCCAGCMGCCGCGGCGGCGGCGGTAA) and r806 (GGACTACVSGGGTATCTAAT) [15]. The study was carried out using the equipment of the resource center “Genomic Technologies, Proteomics and Cell Biology” of ARRIAM.

The general processing of sequences was carried out in the dada2 (v1.14.1) [16] and phyloseq (v1.30.0) [17] packages, according to the recommendations of the authors. The main steps were (i) quality filtering and merging, (ii) de-novo ASV picking, (iii) removing of chimeras, (iv) taxonomic annotation, and (v) reference tree construction. As a training data set for the taxonomic annotation, the SILVA (v138) [18] reference database was used. The main analysis (alpha and beta diversity, heatmaps, etc.) of the results was carried out using the phyloseq and tidyverse (2.0.0) [19] packages in R (v4.3.0). PERMANOVA analysis was performed with vegan (2.5–6) [20]. The search for the closest ASVs was conducted using pairwise alignment of the ASV reference sequences. The closest sequence for every target ASV was determined according to the minimal p-distance in the alignment.

## 3. Results

After primary bioinformatic processing, a total of 2,123,593 reads were selected for 96 samples. The mean depth of sequencing was 22,120 reads per sample and the minimal depth was 10,002 reads per sample. For an alpha diversity analysis, all the data was rarefied to the minimal amount of reads—10,000 reads per sample. Raw reads are available at SRA and available using the PRJNA1098577 identifier, or via direct link: https://www.ncbi.nlm.nih.gov/bioproject/PRJNA1098577 (accessed on 6 November 2024).

### 3.1. Training of Primers (Phase I)

The differences between the microbiomes of the samples from both phases of training were assessed using beta diversity and visualized on the PCoA plot (Figure 3). Pre-cultivation samples from both soils (DF and PS) formed a close cluster, which indicates similarity in terms of abundances of major taxa. In contrast with soils, the diversity of the pre-incubation substrates—oat straw (Oat) and hemp straw (Hemp)—had highly specific microbiota. Trained (post-incubation) communities from Phase II were close to the soil cluster, but the similarity level was different for different primers. Specifically, the PS_Oat samples were extremely close to the soil cluster, whereas DF_Hemp samples had their own cluster far from the soil one. Overall, the microbiomes of trained primers were more affected by soil than by the substrate.

The diversity within microbiomes was measured by the number of observed ASVs, which is one of the alpha diversity indices (Figure 4). According to its value, soils had the most diverse microbial communities, whereas substrates were very low in diversity. Primer communities’ richness was closer to the richness of soils. According to the two-way ANOVA analysis, both factors (soil and substrate type) and their interaction were significant predictors for the number of observed ASVs of the primer community (F_soil_ = 534, p_soil_ < 0.01, F_sub_ = 119, p_sub_ < 0.01, F_soil:sub_ = 45, p_soil:sub_ < 0.01). Thus, alpha diversity also shows the succession between soil and primer microbiomes.

In the circumstances of artificially trained communities, the microbial composition of a specific primer should be the result of an intersection between soil and substrate communities. In a search for origins of the primer microbiota, all of a primer’s taxa were classified according to their presence/absence in soil, source, and primer (Figure 5). “Common” taxa were present in all three types of communities: soil, substrate, and their primer. Taxa were marked as “From soil” or “From substrate” if they were present in primer and soil or primer and substrate, respectively.

Despite our assumption that the microbial composition of a specific primer is the result of an intersection between soil and substrate communities, the analysis revealed many unique ASVs that were present in the primer community in large numbers but were not detected in either the soil or the substrate. The number of these ASVs (marked as Unique) and their abundance are shown in Figure 5. Most of these unique ones are minor ASVs (double- or triple-tones)—but there are several major taxa. These major taxa take up to 1/7 (except PS_Hemp) of all microorganisms in the microbial community of primers (Figure 5).

It is reasonable to suggest that these “unique” ASVs were a result of a bias in sequencing or data processing. To investigate this hypothesis, for every “unique” ASV from a primer, the closest ASV from the soil or substrate was found (see Section 2 Materials and Methods). A range of distance metrics between these two references is shown in Table 1. According to this data, the unique ASVs in most cases had highly similar sequences in soils or substrates. Most ASVs (more than 3/4) are closely connected with soil communities, which is in full concordance with the other results.

### 3.2. Testing of Primers (Phase II)

The cellulolytic activity of primers was tested on both oat straw and hemp straw. The results of measurements of the nutritional parameters (humidity, ash content, decomposition level, nitrate, and carbon levels) of composts had high levels of internal correlation (more than 0.7), therefore we use the decomposition level (according to the weight loss) as the most informative and valuable factor (Figure 6). According to the pairwise Wilcoxon test, applying any of the primers significantly increases the decomposition of substrates.

According to the beta diversity of primers and testing communities (Bray distances), the addition of the primer moderately changed the microbiological composition of the testing communities, without any strict clusters. Moreover, while primers tended to form their specific microbiota, every substrate had its own strict cluster, almost regardless of the primers (Figure 7). The microbial communities of primers are in their own strict clusters, whereas the microbiomes of the test communities (substrates after primer exposure) tend to have their own relaxed clusters according to the substrate type.

## 4. Discussion

### 4.1. Training of Primers (Phase I)

Both soils are highly diverse and show their specific community composition. Podzol soil samples had the highest amount of ASVs, which is typical for a comparison of different soils. The authors of a similar study of two contrasting soils suggested that high microbial diversity corresponds with nutritionally poor soil, whereas low diversity is connected with rich soils [12]. In contrast to soils, both substrates have extremely low diversity. In detail, the hemp straw richness was significantly lower than that of the oat straw. Both the low richness and its variation have also been shown previously [21], and can be connected to the complex structure of both substrates and the limited source of nutrients for microorganisms.

The diversity of trained microbial communities is higher, which is also typical for decomposition processes [22,23]. Variations in the richness of both soils and substrates shape the variations in the richness of primers. The main factor is soil type: highly diverse Podzol soil (PS) corresponds to highly diverse primers (PS_Cannb, PS_Oat), whereas less diverse dark forest soil (DF) leads to less diverse primers (PS_Cannb, PS_Oat). The substrate is less clear but still significant: highly diverse oat substrate leads to highly diverse oat-based primers (PS_Oat, DF_Oat), while less diverse hemp substrate leads to less diverse hemp-based primers (PS_Hemp, DF_Hemp).

According to the results, the microbial composition of trained microbial communities is recruited mostly from soils, reflecting its aspects and properties. A lot of the unique taxa in microbial communities of trained communities (primers) can be explained by relatively low sequencing depth [24], PCR, or even sequencing bias [25]. As a result of training, these microbial communities potentially contain specific, trained microbiota, enriched with cellulolytic microorganisms. But, are they specific to their substrate? Are they equally effective? To answer these questions, we have to test them.

### 4.2. Testing of Primers (Phase II)

All trained microbial communities have demonstrated their efficiency when tested against both available substrates. Oat straw is an easy-to-decompose substrate, and microbial consortia from all primers decompose it easily without any difficulties. More complex hemp straw is not as easy to decompose. It has been shown that the degradation of lignin-enriched cellulose is highly limited. Because a certain lignin-cellulose ratio cannot be exceeded, the highly complex process of lignin degradation limits the process [26]. In the meantime, bacteria show a lower lignin degradation capacity compared to fungi [27]. Fungal communities, in turn, are more conservative and adapt more slowly to environmental changes [28], which results in a slowing down of all degradation processes. Therefore, the overall low level of decomposition in hemp straw is reasonable.

Against our expectations, oat-based primers decompose hemp straw significantly better than hemp-based primers. The reason for this is not clear. Our suggestion is connected with the different decomposition rates for different substrates. Oat straw can be decomposed rapidly. In a dedicated time period, the microbial community processes a high amount of substrate, which leads to a high availability of nutrients, and, as a result, active changes in microbiomes. In contrast, the hemp straw is stiff and difficult to decompose, which in the same time period results in the decomposition of a limited amount of substrate. This leads to a moderate availability of nutrients and little changes in microbial composition. At the end of the training period, oat straw primers turned out to be more adapted for the process of decomposition—more trained—than hemp straw ones.

Future analysis requires investigating by employing both microbial and fungal analysis alongside the functional annotation of communities. It is well known that cellulose degradation recruits specific microorganisms with their specific set of enzymes [29]. More thorough research of these communities, their functional profiles, and enzymes will lead us to a better understanding of cellulose degradation processes. This knowledge will allow humanity to find the most efficient way to recycle complex plant residues into fertilizer or biofuels [30].

## 5. Conclusions

Soils (the source of cellulose-degrading associations) and plant residues (selective media) can be used for the creation of an effective microbial community for cellulose-degradation processes. The taxonomic composition of these communities differs from that of the initial soil, but almost all potential decomposers are recruited from the soil (not from the substrate). In contrast, the cellulose degradation efficiency of these communities depends on the substrate but not on the soil type. There was no substrate specificity, so it is reasonable to use an easily degradable substrate. Similar primer compositions can be used in the future as an easy-to-use accelerator of cellulose decomposition processes.

## Figures and Tables

**Figure 1 microorganisms-12-02276-f001:**
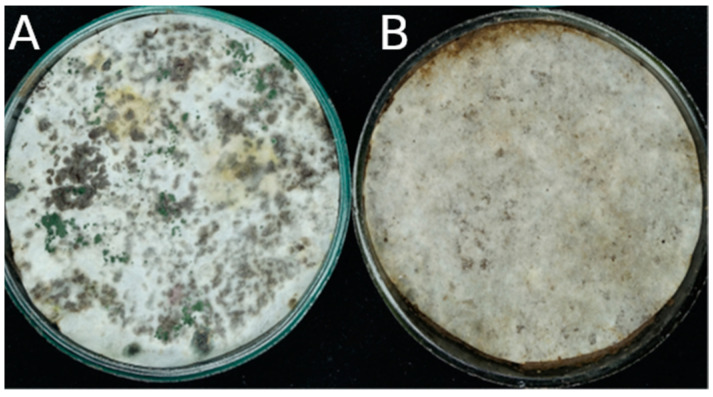
Express-test (degradation of a paper filter on a Petri dish) of the initial cellulolytic activity of soils. (**A**), soil of Chernevaya taiga; (**B**), sod-podzolic soil.

**Figure 2 microorganisms-12-02276-f002:**
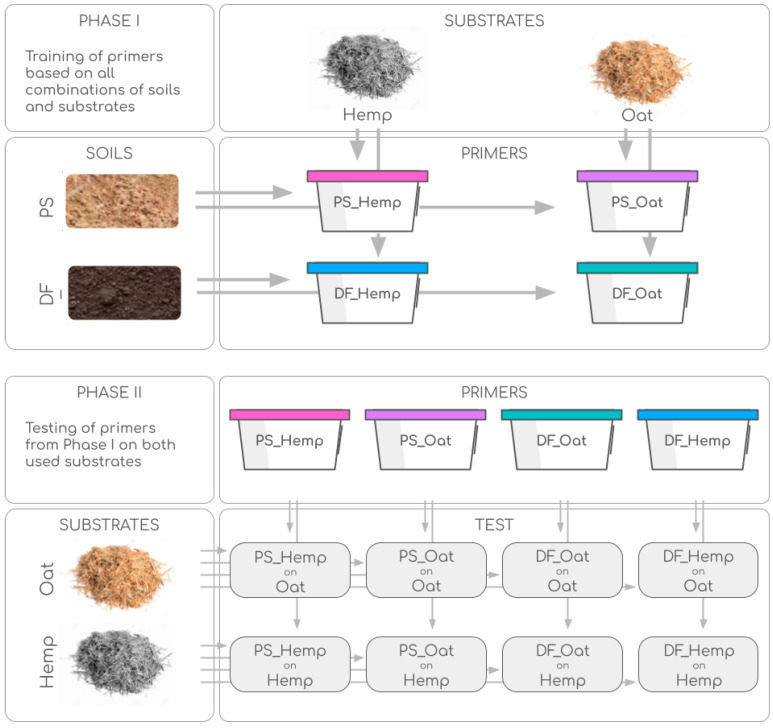
The design of the experiment.

**Figure 3 microorganisms-12-02276-f003:**
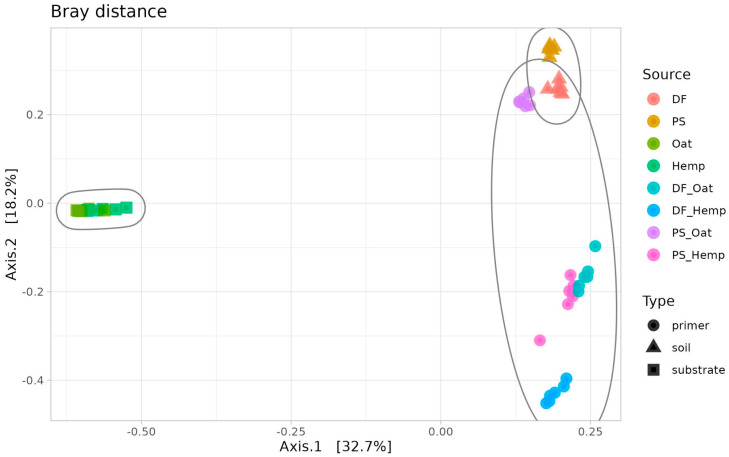
Training of the primers. PCoA ordination of Bray distances in soils, substrates, and primers. Ellipses surround the samples of one type—soil, cellulosic substrate, or primer.

**Figure 4 microorganisms-12-02276-f004:**
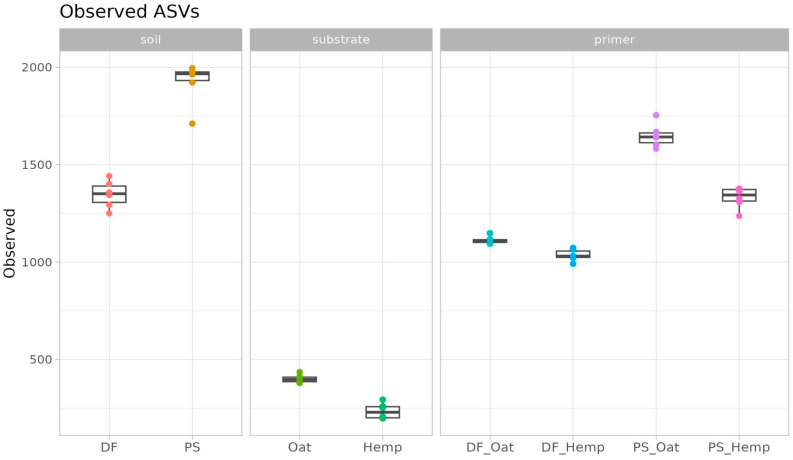
Training of primers. Observed ASVs index in soils, substrates, and primers.

**Figure 5 microorganisms-12-02276-f005:**
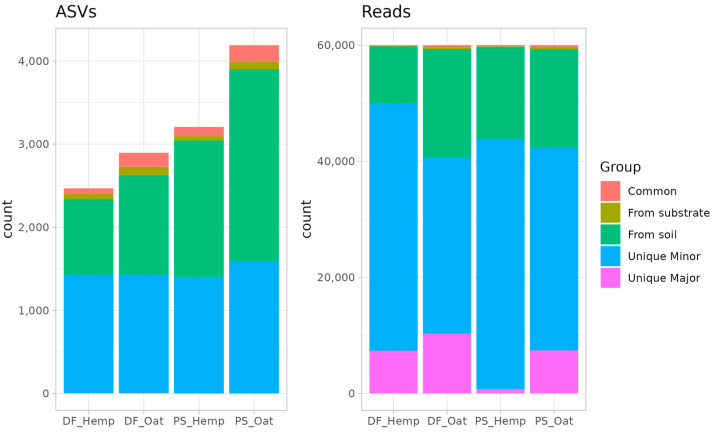
Training of primers. Microbial composition of primers—the amount of ASVs and their abundances. The group represents the origin of the taxa (major taxa assumes abundance > 600 reads).

**Figure 6 microorganisms-12-02276-f006:**
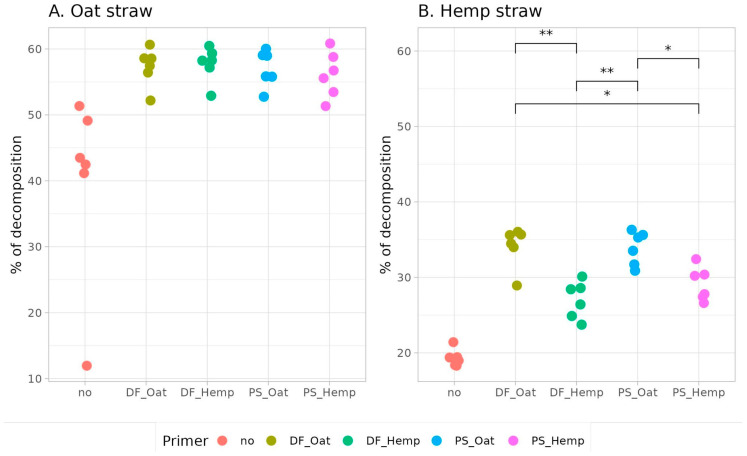
Different substrate decomposition levels (according to the weight loss) of primers (significant pairwise difference is marked by brackets, significance for samples without primer addition (labeled “no”) are not shown, see the text. Significance level: **—*p* < 0.01, *—*p* < 0.05).

**Figure 7 microorganisms-12-02276-f007:**
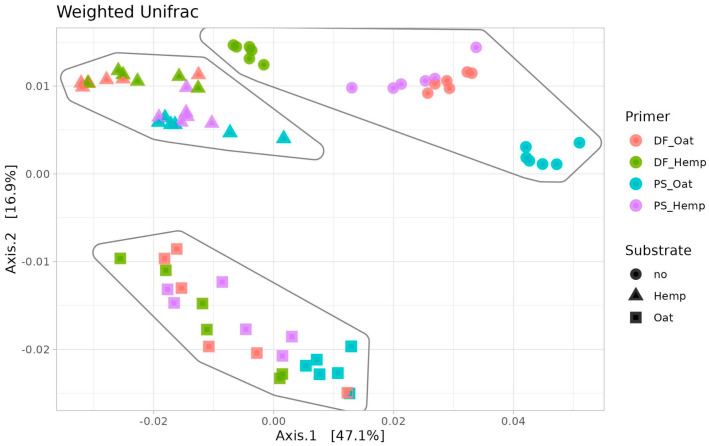
Beta diversity: testing of primers. PCoA ordination of Bray distances of microbial communities. Ellipses surround the samples of one type—primers and substrates after primer exposure.

**Table 1 microorganisms-12-02276-t001:** “Unique” ASVs from primers and their closest ASVs from soil or substrate.

Primer Name	The Possible Source of ASV	Number of Reads	Distances		
			Mean	Q3	Max
DF_Hemp	Common	51	0	0.04	0.27
	Soil	1299	0.02	0.04	0.53
	Substrate	77	0.02	0.06	0.31
DF_Oat	Common	39	0	0.04	0.06
	Soil	1277	0.02	0.06	0.4
	Substrate	113	0.02	0.02	0.26
PS_Hemp	Common	33	0.02	0.04	0.21
	Soil	1328	0.02	0.06	0.51
	Substrate	42	0	0.04	0.17
PS_Oat	Common	44	0.02	0.05	0.15
	Soil	1472	0.02	0.06	0.4
	Substrate	78	0	0.04	0.37

## Data Availability

Raw sequence reads are available at the Sequence Read Archive (SRA) and available using PRJNA1098577 identifier, or via direct link: https://www.ncbi.nlm.nih.gov/bioproject/PRJNA1098577 (accessed 6 November 2024).

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
