# Peer review of "Creation of Cellulolytic Communities of Soil Microorganisms—A Search for Optimal Approaches"

_microorganisms, 2024, doi:10.3390/microorganisms12112276_

Round 1
Reviewer 1 Report
Comments and Suggestions for Authors
Dear Authors,
The manuscript under review, titled "Creation of Cellulolytic Communities of Soil Microorganisms: Search for Optimal Approaches," presents significant research on enhancing soil microbial communities for improved cellulose degradation, a critical process in sustainable agriculture and waste management. The study demonstrates that pre-trained microbial communities, independent of soil origin, increase the efficiency of cellulose breakdown, offering valuable insights for optimizing waste utilization and promoting soil health.
The most important result obtained is that pre-trained microbial consortia (PTMCs) from different soils were equally effective in cellulose degradation. Surprisingly, the consortia trained on oat straw showed higher efficiency in decomposing hemp straw compared to those trained on hemp. The most significant conclusion is that soil type does not influence the efficiency of microbial communities in cellulose degradation. Additionally, using easily degradable substrates like oat straw during the training process can enhance the effectiveness of microbial communities in breaking down more complex substrates, such as hemp.
However, I have some questions based on what is stated in this manuscript.
The study analyzes the efficiency of cellulolytic microbial communities using various statistical techniques to determine key influencing factors. It reveals that pre-trained microbial communities significantly improve the decomposition of both oat and hemp straw, with oat-trained communities performing unexpectedly better on hemp straw. However, some p-values for diversity indices between substrates are non-significant, and the discussion section would benefit from a more focused explanation of the training effect and substrate-specificity. The discussion could be condensed by emphasizing statistically significant trends and providing deeper insights into why oat-trained communities outperform on hemp (e.g., higher initial microbial activity). Consider clarifying the mechanisms behind the interaction between substrate complexity and microbial training for better comprehension by a broader audience.
The entire manuscript requires substantial revision to improve clarity and coherence. The bibliography must be updated to include only those sources that are directly relevant to the study. In the results and discussion sections, focus should be placed solely on the most significant findings, while providing clear and concise explanations that help the reader understand the implications of the research. Additionally, the manuscript must adhere strictly to the journal's formatting guidelines, and the English language throughout the text requires thorough revision to ensure accuracy and readability.
I do not support the outright rejection of manuscripts, as I appreciate the hard work and effort that has gone into this research. However, this manuscript is far from being in a suitable form for publication at this time. I am confident that the authors will be able to revise and improve the manuscript to meet the necessary standards for publication. Therefore, I recommend that they undertake the substantial revisions outlined, ensuring that the manuscript is clear, coherent, and adheres to the journal's guidelines. With the right modifications, I believe the authors can present a version that is optimal for publication.
Congratulations to the authors for their commendable work on this subject.

The English could be improved to more clearly express the research.
Author Response
Reviewer: The study analyzes the efficiency of cellulolytic microbial communities using various statistical techniques to determine key influencing factors. It reveals that pre-trained microbial communities significantly improve the decomposition of both oat and hemp straw, with oat-trained communities performing unexpectedly better on hemp straw. However, some p-values for diversity indices between substrates are non-significant, and the discussion section would benefit from a more focused explanation of the training effect and substrate-specificity.
Authors: We provide more information according to the diversity of both the substrates themselves (lines) and the testing of trained communities. We also provide a dedicated discussion section and discuss our suggestions on the training effect.
Reviewer: The discussion could be condensed by emphasizing statistically significant trends and providing deeper insights into why oat-trained communities outperform on hemp (e.g., higher initial microbial activity).
Authors: The discussion has been substantially reworked. We still do not have clear reason why there is such outperformance, but we discuss possible reasons in Discussion section.
Reviewer: Consider clarifying the mechanisms behind the interaction between substrate complexity and microbial training for better comprehension by a broader audience.
Authors: It is a reasonable proposal, but it requires a lot of effort. During this work, we focused on estimating the effect of pre-trained microbial communities and changing their composition. In our manuscript, we demonstrated the efficiency of this approach and revealed an outperformance effect. The exact mechanisms of this effect need to be investigated in the future using functional analysis and large-scale metagenomic analysis.
Reviewer: The entire manuscript requires substantial revision to improve clarity and coherence. The bibliography must be updated to include only those sources that are directly relevant to the study. In the results and discussion sections, focus should be placed solely on the most significant findings, while providing clear and concise explanations that help the reader understand the implications of the research. Additionally, the manuscript must adhere strictly to the journal's formatting guidelines, and the English language throughout the text requires thorough revision to ensure accuracy and readability.
Authors: We appreciate your substantive comments. We thoroughly revised the entire text of the manuscript, provided a scheme of the experiment, step-by-step clarification of the results, and reorganized the discussion into a separate chapter. We also compared our results with recent studies and formatted the manuscript according to the journal's rules.
Reviewer: I do not support the outright rejection of manuscripts, as I appreciate the hard work and effort that has gone into this research. However, this manuscript is far from being in a suitable form for publication at this time. I am confident that the authors will be able to revise and improve the manuscript to meet the necessary standards for publication. Therefore, I recommend that they undertake the substantial revisions outlined, ensuring that the manuscript is clear, coherent, and adheres to the journal's guidelines. With the right modifications, I believe the authors can present a version that is optimal for publication.
Congratulations to the authors for their commendable work on this subject.
Authors: We would like to thank the reviewer for his thorough evaluation of our manuscript. All suggestions have been accounted for.
Reviewer 2 Report
Comments and Suggestions for Authors
Microorganisms
Manuscript Draft
Manuscript Number: 3269201
Title: Creation of cellulolytic communities of soil microorganisms - search for optimal approaches
Article Type: Research article
General Comments on MDPI Questions that Reviewers must answer:
- Is the manuscript clear, relevant for the field and presented in a well-structured manner?
The manuscript is relevant to the field since it investigates cellulolytic microbial communities associated with oat straw and hemp shives across two different types of soil in Russia. However, the manuscript needs to be improved as a communication type manuscript in the following areas before it is suitable for publication:
1) Please clarify in the Introduction section why the research is importance when it comes to better investigation of cellulolytic soil microorganisms. For example, can these microbes be used to better break down cellulose of oat and hemp feed stocks for production of bio-fuels? What is the application(s) that benefits humanity?
2) The 3. Results and Discussion section is just the presentation of results and there is no substantive discussion. Please add a separate 4. Discussion section where you contrast the results to past literature as well as discuss the implications of the results (e.g., what is/are the implication(s) for benefiting humanity?).
3) There is an overuse of abbreviations. Please only use abbreviations if there is excessive repetition of the term being abbreviated. Please do NOT use abbreviations in the Abstract, Figures, Tables, and captions for figures and tables.
4) Please double the number of cited References. Even for a communication type manuscript, the number of citations is too low.
5) Please refer back to the MDPI Microorganisms Word template and include all Back Matter sections located between the Conclusion and References. The formatting needs to be correct. Also, the formatting of the References need to follow the exact format provided in the MDPI Microorganisms Word template. Please also see my specific comments below under “*Other edits that need to be done”.
- Are the cited references mostly recent publications (within the last 5 years) and relevant? Does it include an excessive number of self-citations?
Only 7 of the 16 cited references have been published within the last 5 years. Please increase the number of references published since 2019. All references are relevant to the research topic and there are not excessive self-citations.
- Is the manuscript scientifically sound and is the experimental design appropriate to test the hypothesis?
The manuscript is scientifically sound and experimental methods are appropriate. The goal and objectives of the research need to be more clearly stated in the last paragraph to the Introduction section. Please begin the paragraph by stating what the overall goal of the research is and please follow this by listing the objective(s) of the research.
- Are the manuscript’s results reproducible based on the details given in the methods section?
The manuscript’s results are reproducible by others based on what is described in 2. Materials and Methods in order to repeat the experiment. Please capitalize the Methods in the section header.
- Are the figures/tables/images/schemes appropriate? Do they properly show the data? Are they easy to interpret and understand? Is the data interpreted appropriately and consistently throughout the manuscript? Please include details regarding the statistical analysis or data acquired from specific databases.
In general, the figure and tables are fine with the exception of not using abbreviations in the tables as well as captions for tables and figures.
- Are the conclusions consistent with the evidence and arguments presented?
The Conclusions section which is OK to just be one or two paragraph(s) needs to be written in paragraph form and then end with a couple of sentences on how future research can improve upon the current work.
- Please evaluate the data availability statements to ensure it is adequate.
Please add the Data Availability Statement. Please add the Ethics Statement where it is stated as not applicable since the experiment did not involve animals nor human subjects.
*Other edits that need to be done:
1) All major words in the title are capitalized as required by MDPI:
Creation of Cellulolytic Communities of Soil Microorganisms: Search for Optimal Approaches
2) The Abstract should one paragraph.
3) The first keyword needs to be capitalized. All other keywords should not be capitalized unless the word is naturally capitalized. The keywords need to be in alphabetical order.
4) The format for citations in the writing is [#] and not (#).
5) Paragraphs by definition have a minimum of 3 sentences (1 topics sentence followed by at least 2 supporting sentences). Please make sure there are no 1 sentence nor 2 sentence paragraphs.
6) Please make sure to write out Figure and Not as Fig. or figure.
7) The citation format of (Callahan et al., 2016) on L133-134 is not correct so please delete and just use [13].
Author Response
Dear reviewer, thank you for your extensive work. We appreciate your substantive comments. We thoroughly revised the entire text of the manuscript, provided a scheme of the experiment, step-by-step clarification of the results, and reorganized the discussion into a separate chapter. Here are the answers for the questions:
Q: Please clarify in the Introduction section why the research is importance when it comes to better investigation of cellulolytic soil microorganisms. For example, can these microbes be used to better break down cellulose of oat and hemp feed stocks for production of bio-fuels? What is the application(s) that benefits humanity?
A: We provide this information – lines 87-95
Q: Results and Discussion section is just the presentation of results and there is no substantive discussion. Please add a separate 4. Discussion section where you contrast the results to past literature as well as discuss the implications of the results (e.g., what is/are the implication(s) for benefiting humanity?).
A: The Discussion section has been added, providing hypothesis and comparisons with current knowlege
Q: There is an overuse of abbreviations. Please only use abbreviations if there is excessive repetition of the term being abbreviated. Please do NOT use abbreviations in the Abstract, Figures, Tables, and captions for figures and tables
A:We use only one abbreviation (pre-trained microbial community – PTMC). But it can be replaced by another term – we have replaced it, and use the term “primer”.
Q: Please double the number of cited References. Even for a communication type manuscript, the number of citations is too low.
A: We double the number of cited manuscripts
Q: Please refer back to the MDPI Microorganisms Word template and include all Back Matter sections located between the Conclusion and References. The formatting needs to be correct. Also, the formatting of the References need to follow the exact format provided in the MDPI Microorganisms Word template
A: We use MDPI Microorganisms template and Zotero reference manager for the formatting of references
Q: Only 7 of the 16 cited references have been published within the last 5 years. Please increase the number of references published since 2019.
A: We have revised the manuscript using recomended articles in both Intro and Discussion sections. In Materials and Methods we kept references, wich were recommended by the authors’ of packages and tools.
Q: The goal and objectives of the research need to be more clearly stated in the last paragraph to the Introduction section. Please begin the paragraph by stating what the overall goal of the research is and please follow this by listing the objective(s) of the research.
A: We clarify the aims in the last paragraph of Introduction – lines 96-103
Q: The Conclusions section which is OK to just be one or two paragraph(s) needs to be written in paragraph form and then end with a couple of sentences on how future research can improve upon the current work.
A: We replace the Conclusion according to your recommendations
Q: Please add the Data Availability Statement. Please add the Ethics Statement where it is stated as not applicable since the experiment did not involve animals nor human subjects.
A: Data Avaliability Statement and Ethics Statement have been added – lines 315-318
Q: Other edits that have to be done
1) All major words in the title are capitalized as required by MDPI:
Creation of Cellulolytic Communities of Soil Microorganisms: Search for Optimal Approaches
2) The Abstract should be one paragraph.
3) The first keyword needs to be capitalized. All other keywords should not be capitalized unless the word is naturally capitalized. The keywords need to be in alphabetical order.
4) The format for citations in the writing is [#] and not (#).
5) Paragraphs by definition have a minimum of 3 sentences (1 topics sentence followed by at least 2 supporting sentences). Please make sure there are no 1 sentence nor 2 sentence paragraphs.
6) Please make sure to write out Figure and Not as Fig. or figure.
7) The citation format of (Callahan et al., 2016) on L133-134 is not correct so please delete and just use [13].
A: All recommendations are very helpful, we followed all of them
Round 2
Reviewer 1 Report
Comments and Suggestions for Authors
Dear Authors,
While I appreciate the revisions made to the manuscript, it remains unclear exactly what specific changes have been implemented by the authors. Simply renaming the subheadings does not necessarily indicate a comprehensive review and revision of the content. To facilitate a more thorough evaluation, could you please highlight the specific modifications made using a different font color or other visual cues? This would greatly enhance the clarity of the revisions and allow for a more accurate assessment of the overall improvements.

Dear Editors,
The authors have not adequately addressed the feedback provided. Simply rewriting the subheadings does not constitute a thorough revision of the manuscript.
Reviewer 2 Report
Comments and Suggestions for Authors
Thanks for making requested edits. Only 2 minor types of edits:
1) Please make sure all paragraphs have at least 3 sentences. For example on L245-247, please add another supporting sentence to complete the paragraph.
2) On L276 for example, please capitalize Primers as all major words in section and sub-section headers require capitalization.
Thanks!
Author Response
Thanks again for the important notes
Reviewer: Please make sure all paragraphs have at least 3 sentences. For example on L245-247, please add another supporting sentence to complete the paragraph.
Authors: The manuscript was checked again, now all paragraphs contain 3 sentences or more. In case of the exact L245-247, we have added it to the previous paragraph.
Reviewer: On L276 for example, please capitalize Primers as all major words in section and sub-section headers require capitalization.
Authors: We capitalized major words in sections and sub-sections according to your advice
Round 3
Reviewer 1 Report
Comments and Suggestions for Authors
Dear Authors,
I have reviewed the revised manuscript and note the significant improvements made by the authors in response to the feedback provided. I concur with the publication of the manuscript in its current form.
